# Performance Characteristics in Runner of an Impulse Water Turbine with Splitter Blade

**Lingdi Tang** [1,2,*], **Shouqi Yuan** [1], **Yue Tang** [1] **and Zhijun Gao** [3]

1   Research Center of Fluid Machinery Engineering and Technology, Jiangsu University,
    Zhenjiang 212013, China; shouqiy@ujs.edu.cn (S.Y.); tomt@ujs.edu.cn (Y.T.)
2   Key Laboratory of Water-Saving Irrigation Engineering, Ministry of Agriculture, Xinxiang 453000, China
3   Jiangsu Wanagda Sprinkler Co., Ltd., Changzhou 213299, China; johnzhuang@wangdairrigation.com
*   Correspondence: lingdit@ujs.edu.cn

**Abstract:** The impulse water turbine is a promising energy conversion device that can be used as mechanical power or a micro hydro generator, and its application can effectively ease the current energy crisis. This paper aims to clarify the mechanism of liquid acting on runner blades, the hydraulic performance, and energy conversion characteristics in the runner domain of an impulse water turbine with a splitter blade by using experimental tests and numerical simulations. The runner was divided into seven areas along the flow direction, and the power variation in the runner domain was analyzed to reflect its energy conversion characteristics. The obtained results indicate that the critical area of the runner for doing the work is in the front half of the blades, while the rear area of the blades does relatively little work and even consumes the mechanical energy of the runner to produce negative work. The high energy area is concentrated in the flow passage facing the nozzle. The energy is gradually evenly distributed from the runner inlet to the runner outlet, and the negative energy caused by flow separation with high probability is gradually reduced. The clarification of the energy conversion performance is of great significance to improve the design of impulse water turbines.

**Keywords:** impulse water turbine; runner domain; energy conversion; flow characteristics



## 1. Introduction

With the growing demand for energy, the development and utilization of environmentally friendly renewable energy has received more and more attention. Actively researching and developing environmentally friendly renewable energy technologies and gradually abandoning unsustainable development models have become important ways to alleviate energy shortages, reduce environmental pollution, and achieve harmonious development of energy and environment. Hydropower is considered an environmentally friendly renewable energy with substantial development potential [1]. A water turbine is an energy recovery or utilization device that uses water pressure to rotate the runner and converts water pressure energy into turbine mechanical energy. It has widespread application in many fields, such as water conservancy, petrochemical, seawater desalination, and mechanical power. It also has a positive effect on reducing energy consumption per unit of output [2–5].

In the last few years, the entire network with the water turbine has been considered a unit, and unified modeling has been used to analyze energy conversion of the turbine. Ma et al. analyzed the output power of the whole system by formulating a MINLP problem to evaluate energy conversion of a hydro turbine and presented that this hydro turbine has greater energy-saving potential [6]. Nagode et al. controlled the turbine power through an internal model control on the strength of a turbine fuzzy model to ensure better energy conversion for a Francis turbine [7]. Riglin et al. analyzed the relationship between the output power changes in a propeller-based hydrokinetic turbine and the

system components, including the generator, transmission, gearbox, etc. [8]. Du et al. designed a pump as turbine (PAT) control strategy for the water supply system and investigated the output power changing trend with operating time. It was found that the power output experienced an increase and then stayed stable at a certain value after a slight fluctuation [9].

Other researchers analyze the influence of different components and their different design parameters on energy conversion through experiment and numerical analysis. Analyzing different components, such as blade number, air suction hole, ducted nozzle, deflector plate, and others, have been considered. A reasonable increase in the number of blades can increase the output power, but it will also have a negative impact on the turbine flow. Hung et al. observed that the water wheel turbine with more blades can contribute to the useful torque, but the main flow from the inlet was obviously obstructed [10]. Nishi et al. also revealed that the cross-flow water turbine runners with more blades has a relatively large peak torque, but the torque fluctuation was fairly high [11]. The air suction hole with air supplying can reduce the negative torque considerably and then significantly improve the output power of the cross-flow turbine [12]. The ducted nozzle for the Savonius water turbine will make the flow direction perpendicular to the advancing blade, impede the flow from flowing to the returning blade, and then cut down the negative torque acting on the returning blade [13]. A properly placed deflector plate can affect the flow distribution and largely increase the obtained power of the turbine by noticeably increasing the flow area. Kerikous et al. presented that when the deflector plate merely covers the half-returning blade on the outside and is away from the turbine, it can apparently be preferable for the output power performance [14]. Golecha et al. concluded that the deflector plate placed upstream to the fluid flow will obstruct the flow coming towards the returning blade, reduce the negative or reverse torque on the returning blade, and then significantly improve the output power characteristics of the Savonius rotor [15]. Sritram et al. observed that the deflector plates for the water turbine can be helpful to retain the right amount of water and yield higher torque [16]. Prasetyo et al. also draw the conclusion that the deflector for the Savonius water turbine can set the water towards the concave blades and prevent the water towards the convex blades. It will enhance the output torque and produce larger output electrical power of the turbine [17].

Analyzing different design parameters of flow passage components, such as runner inlet arc angle, attack angle, blade curvature angle, blades aspect ratio, blade curvature radius, blade preset pitch angle, blade twist angle, blade chord length, and others, have been considered [18,19]. Du et al. indicated that a smaller runner inlet arc angle of the cross-flow turbine can raise the flow velocity at the inlet of the runner, increase pressure difference from upstream to downstream in the runner, and then generate greater output power [20]. Li et al. found that the flow angle change at the inlet and outlet of the pump-turbine runner leads to a decrease in the work and a reduction in the Euler momentum [21]. Yang et al. investigated the hydraulic performance for a lift-type hydro turbine and found that the position where the positive peak of the torque appears is in the middle of the upper stream, which is also the phase angle of the positive peak of the attack angle, but theoretically, the blade has no torque output at the negative peak of attack angle [22]. Sanditya et al. showed the result that the increase in the blade curvature angle of the Savonius water turbine is not in direct proportion to the growth in the produced output power, where the blade curvature angle has an optimal point at a specific blade curvature angle [23]. Saleem et al. explored the performance of a gravitational water vortex turbine. It was found that a blade with a larger curvature radius will experience a larger torque and when the blade aspect ratio reaches a larger area in contact with water, the received torque will be larger [24]. Mao et al. investigated the influence of design parameters on the performance of water turbine through numerical analysis. It was found that, with the increase of blade radius, a greater force through the integration of the pressure on the semicircle surface is generated and then a greater torque is produced relative to the center of rotation. With the increase of pitch angle, the force will creates a clockwise torque that is opposite to the rotor

rotation direction and helpful to reduce the net torque on the blades. With the increase of the blade chord length, the negative torque range will become wider [25]. Mosbahi et al. observed that the turbulent kinetic energy is enhanced with the reduction of the blade twist angle for a novel combined water turbine, while the turbulent kinetic energy value reduces with the decrease of the blade twist angle [26].

The literature mentioned above focuses on analyzing the influence of different components on the energy conversion results. There are also several published studies in the literature investigating the energy composition and conversion process in the main working section. Nishi et al. categorized the torque on the runner of the cross flow water turbine by the dynamic component decomposition and conducted torque component analysis. It was found that the gravitational component was dominant at a low flow rate and the angular momentum component was dominant at medium and high flow rates [27]. Yang et al. analyzed the kinetic energy, pressure energy, and total energy at the impeller of the mixed-flow pump and found that the energy at the outlet end of the impeller is dominated by pressure energy, but the distribution of total energy at the outlet is more similar to that of kinetic energy [28]. Zhang et al. showed that the actual flow channel before the throat at the impeller outlet for a centrifugal pump is a key area of energy conversion, where most flow energy is gained. The essential form of energy loss in the impeller is wall friction loss, which is focused on blade suction side of the impeller and close to the outlet of the impeller [29]. Miao et al. showed that the critical areas for fluid to perform work on the runner of the pump as turbine (PAT) are in the front and middle sections. The area behind the impeller blades does relatively little work on the runner at a low flow rate. When the flow rate gradually increases, this area not only does no work on the runner but expends mechanical energy from the runner [30]. Zhang et al. showed that the concave side for the advancing blade of the Savonius hydrokinetic rotor can generate high torque and the downstream area of the advancing blade can produce negative torque [31]. Shi et al. studied the internal flow and performance of the axial flow pump and found that the main part of the pump for energy conversion is the front third of the impeller and the energy conversion capacity of the impeller near the rim is stronger than that of the hub [32,33]. Laín et al. revealed that the torque transferred from the fluid to each blade for the Darrieus water turbine was mainly in the upstream semicircle, while such a transfer was very small in the downstream semicircle, and the peak value reduces with the increase of the rotational speed for each blade [34].

As discussed above, there exists an ambiguity as to how the energy converts inside a water turbine and its characteristics. The impulse water turbine studied in this paper works under pressure and is often used as mechanical power or as a micro hydro generator. The features of the impulse water turbine different to the traditional water turbine. It is required to have sufficient output at the lowest flow rate and low head loss simultaneously, as well as a flat efficiency curve and peak efficiency towards a low flow rate [35,36]. Hence, this paper is designed to evaluate the turbine performance and its flow field by studying the energy conversion process in the runner domain as the main flow section of an impulse water turbine. The results are expected to conduce present knowledge on impulse water turbines and the establishment of higher performance design guidelines for the impulse water turbine.

## 2. Methodology

### 2.1. Physical Model

The present work simulates the whole passage flow of an impulse water turbine, with open runner and inlet nozzle, as shown in Figure 1. Performance parameters of the water turbine are listed in Table 1. The entire computational domain consists of three sub-domains, an inlet static domain containing nozzle, an outlet static domain, and an internal rotation domain containing the runner.

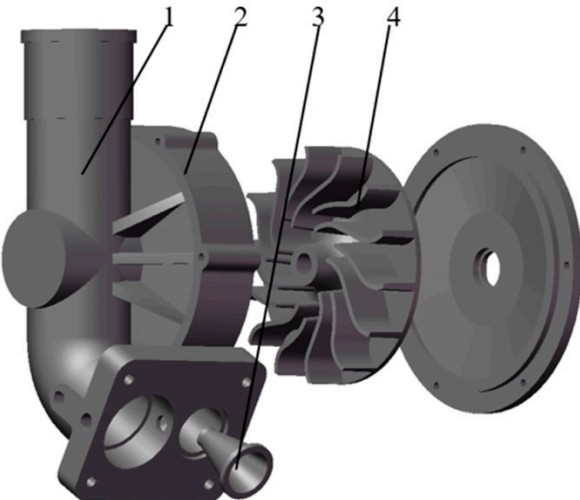

**Figure 1.** Geometry of the impulse water turbine. (**1**) draft tube; (**2**) turbine casing; (**3**) nozzle; (**4**) open runner.

**Table 1.** Specifications of the impulse water turbine.

| Category | Parameter | Value |
|---|---|---|
| Design point | Flow/$Q_d$ (m$^3$/h) | 17.5 |
| | Head/$H_d$ (m) | 10 |
| | Rotational speed/$n_d$ (rpm) | 700 |
| Geometry value | Runner inlet diameter/$D_1$ (mm) | 175 |
| | Runner outlet diameter/$D_2$ (mm) | 55 |
| | Blade inlet angle/$\beta_b$ (°) | 63 |
| | Blade number/$Z_1$ | 14 |
| | Splitter blade number/$Z_2$ | 7 |
| | Runner inlet width/$b_1$ (mm) | 30 |

*2.2. Numerical Method and Schemes*

The fluid internal flow analysis in water turbines needs a solution to the computational domain. The governing equations employed to the computational model contain the Reynolds-averaged continuity, the momentum equation, and the turbulence model. However, there was no widespread turbulence model to meet all flow cases till now, and researchers need to choose a suitable turbulence model according to different phenomena [37,38]. In this paper, the correlation test of the turbulence model is conducted by the utilization of four turbulence models: standard k-ε, RNG k-ε, standard k-ω, and SST k-ω turbulence models. Table 2 gives the comparisons between the numerical and experimental results for these four turbulence models at several flow conditions of $0.7Q_d$, $1.0Q_d$ and $1.3Q_d$. It can be seen that the standard k-ε model shows a reasonable turbine performance compared to other turbulence models. Thus, the standard k-ε model is selected for the following calculations. The continuity equation is:

$$\frac{\partial \rho}{\partial t} + \frac{\partial}{\partial x_j}(\rho u_j) = 0 \tag{1}$$

and the momentum equation becomes:

$$\frac{\partial \rho u_i}{\partial t} + \frac{\partial}{\partial x_j}(\rho u_i u_j) = -\frac{\partial p'}{\partial x_i} + \frac{\partial}{\partial x_j}\left[u_{eff}\left(\frac{\partial u_i}{\partial x_j} + \frac{\partial u_j}{\partial x_i}\right)\right] + S_M \tag{2}$$

$$p' = p + \frac{2}{3}\rho k + \frac{2}{3}\mu_{eff}\frac{\partial u_k}{\partial x_k} \tag{3}$$

where $S_M$ represents the summation of body forces and $\mu_{eff}$ represents the effective viscosity that causes turbulence.

**Table 2.** Results from the numerical calculation and experiment with different turbulent models.

| Turbulence Model | Standard k-ε | RNG k-ε | Standard k-ω | SST k-ω | Test Value |
|---|---|---|---|---|---|
| η at 0.7Q$_d$/% | 38.51 | 38.85 | 41.53 | 41.22 | 34.17 |
| η at 1.0Q$_d$/% | 42.93 | 43.17 | 45.58 | 45.62 | 39.09 |
| η at 1.3Q$_d$/% | 35.75 | 35.97 | 37.79 | 37.28 | 34.12 |

The transport equation for the turbulent kinetic energy and turbulence dissipation rate:

$$\frac{\partial(\rho k)}{\partial t} + \frac{\partial}{\partial x_j}(\rho k u_j) = \frac{\partial}{\partial x_j}\left[\left(\mu + \frac{\mu_t}{\sigma_k}\right)\frac{\partial k}{\partial x_j}\right] + P_k - \rho\varepsilon + P_{kb} \tag{4}$$

$$\frac{\partial(\rho\varepsilon)}{\partial t} + \frac{\partial}{\partial x_j}(\rho u_j\varepsilon) = \frac{\partial}{\partial x_j}\left[\left(\mu + \frac{\mu_t}{\sigma_\varepsilon}\right)\frac{\partial\varepsilon}{\partial x_j}\right] + \frac{\varepsilon}{k}(C_{\varepsilon 1}P_k - C_{\varepsilon 2}\rho\varepsilon + C_{\varepsilon 1}P_{\varepsilon b}) \tag{5}$$

where $C_{\varepsilon 1}$ is 1.44, $C_{\varepsilon 2}$ is 1.92, $\sigma_k$ is 1.0, $\sigma_\varepsilon$ is 1.3, $P_{kb}$ and $P_{\varepsilon b}$ are the influence of buoyancy forces, and $P_k$ represents the turbulence production by reason of the viscous force.

STAR CCM+ is used to generate the polyhedral mesh in the whole computational domain (see Figure 2). To accurately describe the boundary layer flow, prism boundary layer elements are applied to the blade profile. The number of prism layers was 10 with a growth rate of 1.2.

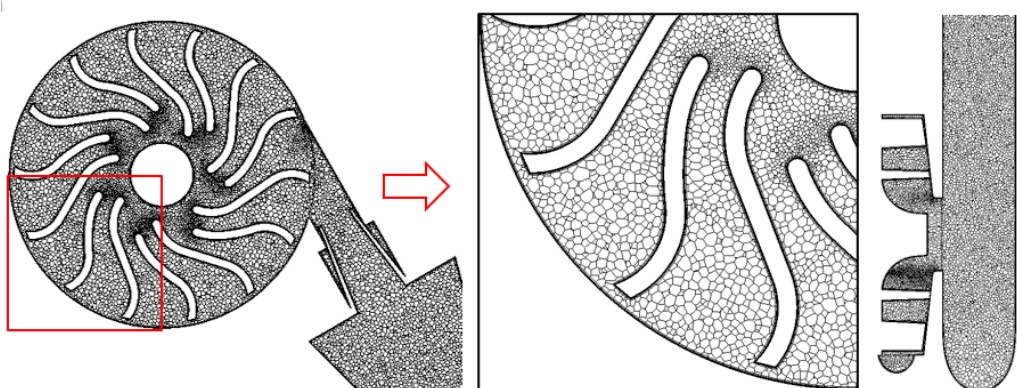

**Figure 2.** Computational grids of the impulse water turbine.

Five grid sizes are considered at the design condition to investigate the grid sensitivity [39,40]. The calculated efficiency and the head with the same numerical settings are shown in Table 3. The results reveal that the efficiency and the head became constant after the grid number was more than $1.25 \times 10^6$. Therefore, the grid number of $1.25 \times 10^6$ was sufficient and was used for subsequent CFD calculation. Moreover, the value of y+ in the first cell next to the solid boundary for each grid was determined to be below 5 for the runner blade passage and below 18 for the other parts of the turbine passage.

**Table 3.** The efficiency and head values with different grid sizes.

| Grid Size/mm | 6 | 4 | 2 | 1 | 0.5 |
|---|---|---|---|---|---|
| Grid numbers | 364515 | 551644 | 1254972 | 2448973 | 4723698 |
| η/% | 44.62 | 43.65 | 42.93 | 42.51 | 42.18 |
| H/m | 10.36 | 10.11 | 9.55 | 9.52 | 9.51 |

The boundary conditions were set as the flow rate at the inlet and total pressure (0.3 MPa) at the outlet, in accordance with the experiment below. No-slip wall conditions were applied for all the walls. The convergence criterion was set to $10^{-5}$. Frozen rotor models were set to rotor-stator interfaces, one of which was between the turbine casing and runner, and the other was between the runner and draft tube.

### 2.3. Experimental Setup

The experiment rig (see Figure 3) was located in Jiangsu Wanagda Sprinkler Co., LTD, Changzhou, China. It was an open experimental system, which consisted of the tested water turbine, water conveying pump, control valve, pipeline, magnetic particle brake, pressure transducer, magnetic flow meter, torque, and speed sensor, etc., as shown in Figure 4. The water conveying pump sucked water from the reservoir and supplied it to the tested water turbine through a pipeline after it was pressurized. The supplying water was controlled by the valve. The flow meter was installed in the pipeline between the outlet of the pump and the inlet of the turbine and was located at the upper stream of the control valve. The water turbine outlet pipe was connected to the reservoir, and the water turbine inlet and outlet pipelines were equipped with a pressure transducer.

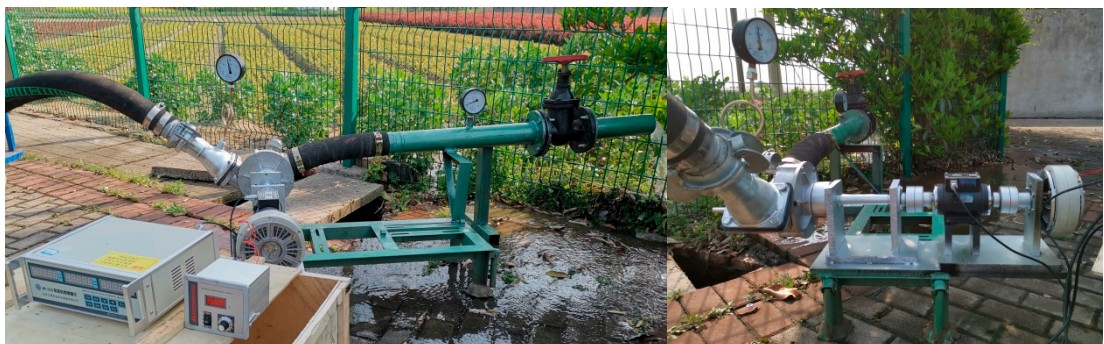

**Figure 3.** Experiment rig.

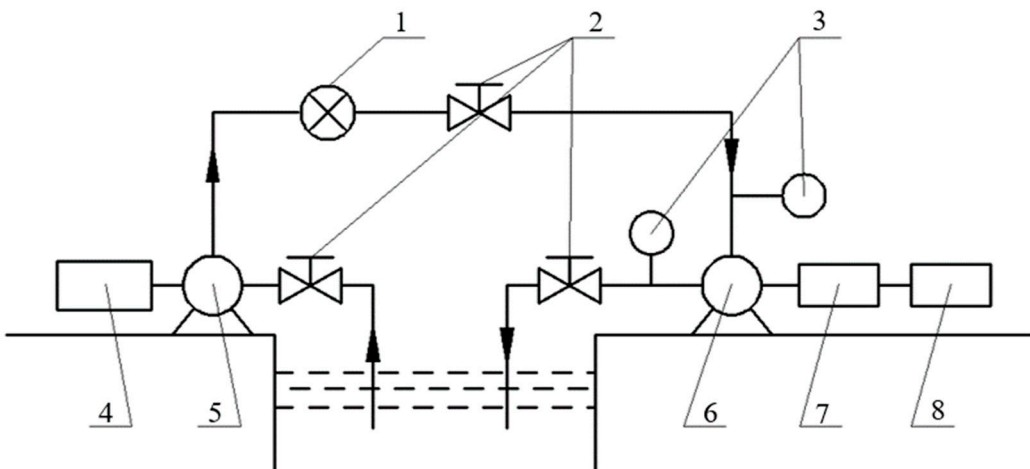

**Figure 4.** Experimental system layout. (**1**) Magnetic flow meter; (**2**) control valve; (**3**) pressure transducer; (**4**) motor; (**5**) water conveying pump; (**6**) tested water turbine; (**7**) torque and speed sensor; (**8**) magnetic particle brake.

## 3. Results and Analysis

### 3.1. Hydraulic Performance

The performance investigation for the impulse water turbine was carried out in the experiment rig, as shown in Figure 3. The overall trends of turbine efficiency in the

experiment were similar in composition to that in the numerical calculation, as shown in Figure 5. The difference between the calculation and experiment results for the efficiency at the design condition was about 9.8%, and the difference at the low flow rate was obviously bigger than that of the high flow rate. These differences may be caused by the simplified geometry, the turbulence model, unconsidered mechanical friction loss, etc. From the above, it can be considered that the qualitative tendency of the turbine performance from calculation results agrees with the experimental result and qualitative analysis can be performed on the basis of the calculation result.

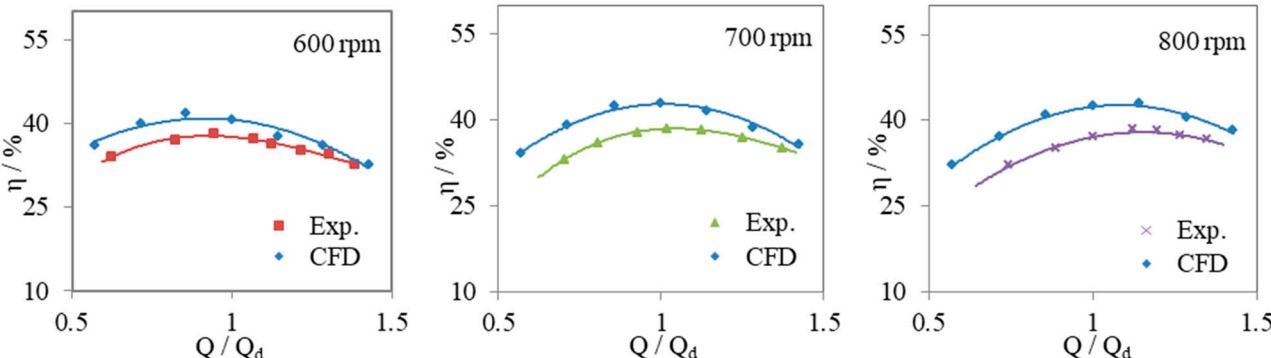

**Figure 5.** The efficiency in the experiment and the numerical calculation.

Steady flow calculations for the water turbine performance at different rotational speeds were performed, as shown in Figure 6. Here, the water turbine output growth with the raise of flow rate was at the same rotational speed. The water turbine efficiency increased as the flow rate increased, to a certain extent, and then reduced as the flow rate increased further. The peak efficiency was 42.93% and it moved toward a higher flow rates as the rotational speed increased. When the turbine ran between 600 rpm and 700 rpm, the high efficiency point was biased towards a low flow rate and the high efficiency area of 700 rpm was wider than that of 600 rpm. In short, the efficiency of this water turbine is not remarkable, btu it can still be further improved, and the high efficiency range is expected to be broadened.

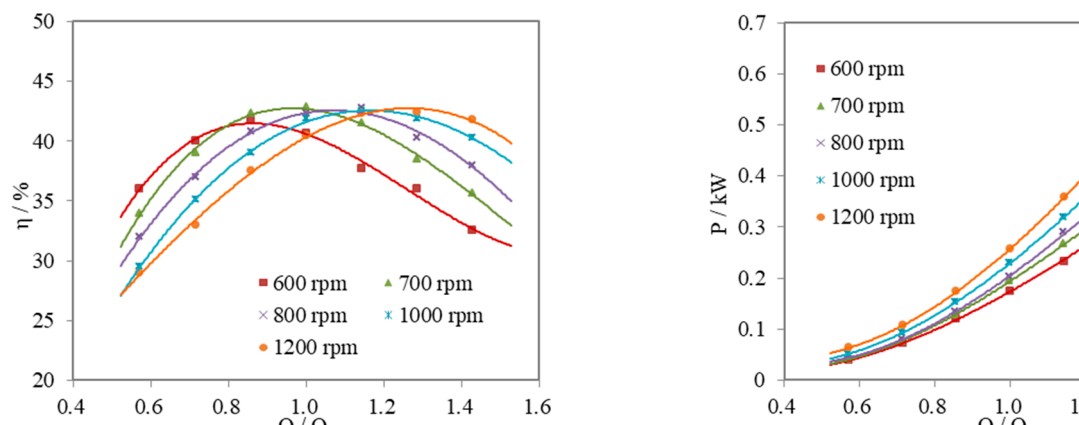

**Figure 6.** Hydraulic performances for the impulse water turbine.

### 3.2. Energy Conversion Performance

For detailed analysis of the energy conversion process in the water turbine runner, the runner was divided into seven parts according to the diameter from the inlet to the outlet. Locations of eight cross-sections used for partitioning are given in Figure 7. Cross-section 1 was at the runner inlet and cross-section 8 was at the runner outlet. The energy conversion

characteristics in each part of the runner can be clarified by analyzing the water turbine runner in the partitioned mode.

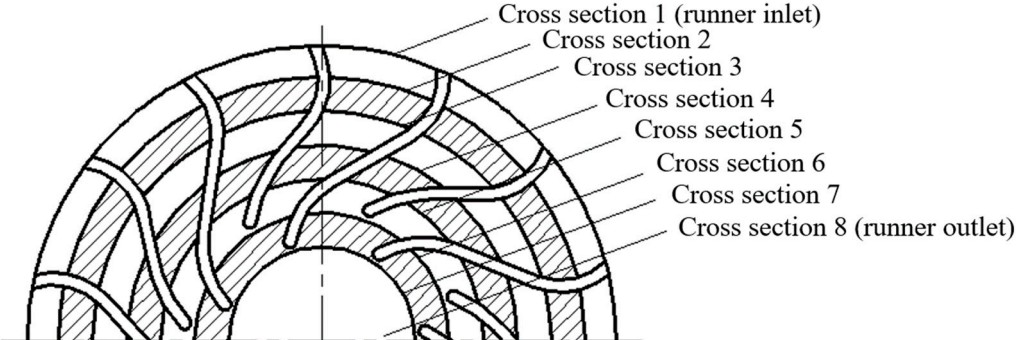

**Figure 7.** Schematic diagram of the runner partition.

Power is the rate of energy change, so here, the power change was used to reflect the energy conversion. Figure 8 gives the power variation for all set cross-sections of the runner along the radius at several working conditions. The power value was specifically calculated according to Equation (6),

$$P_{\mathrm{sec}} = \int_A p_t v \cdot n dA \tag{6}$$

in which the integral term is the power density, $p_t$ is the total pressure, and $v$ is the velocity found from solving the mass conservation equation.

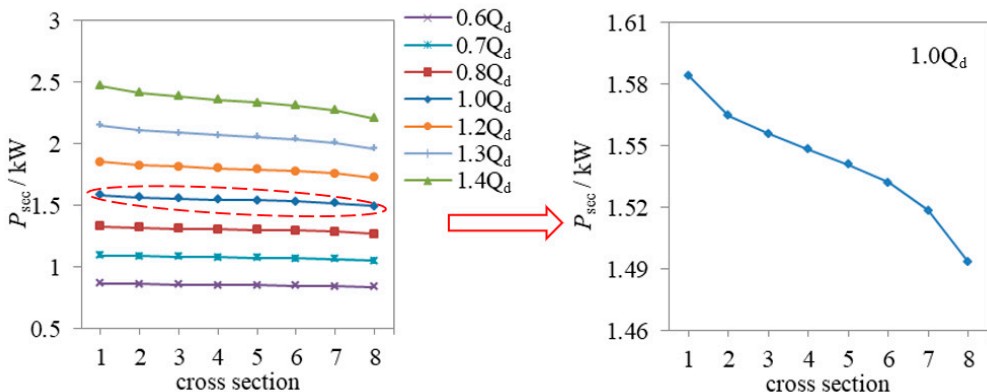

**Figure 8.** Power for each cross-section of the runner.

Figure 8 illustrates that the changing trend of the power for the runner along each cross-section under different flow rates was similar. On the cross-section from the inlet to outlet of the runner, the power showed a downward trend, and the gradient of the power drop when the flow was large was greater than that of the small flow. The power of the runner inlet and outlet dropped faster and the middle part decelerated slightly. It can be considered that the capacity of the runner to do work from the inlet to outlet gradually decreased, the front half of the blade was the main part doing the work, and the capacity of the runner to do work gradually decreased with the increase of flow rate.

The total power included static pressure power and dynamic pressure power, which can be obtained according to Equations (7) and (8), respectively. Figure 9 gives the changes of static pressure power and dynamic pressure power on all set cross-sections of the runner at several flow rates.

$$P_s = \int_A p_s v \cdot n dA \tag{7}$$

$$P_d = \int_A p_d v \cdot n dA \tag{8}$$

in which $P_s$ is static pressure power, $P_d$ is dynamic pressure power, $p_s$ is static pressure, and $p_d$ is dynamic pressure in the absolute coordinate system.

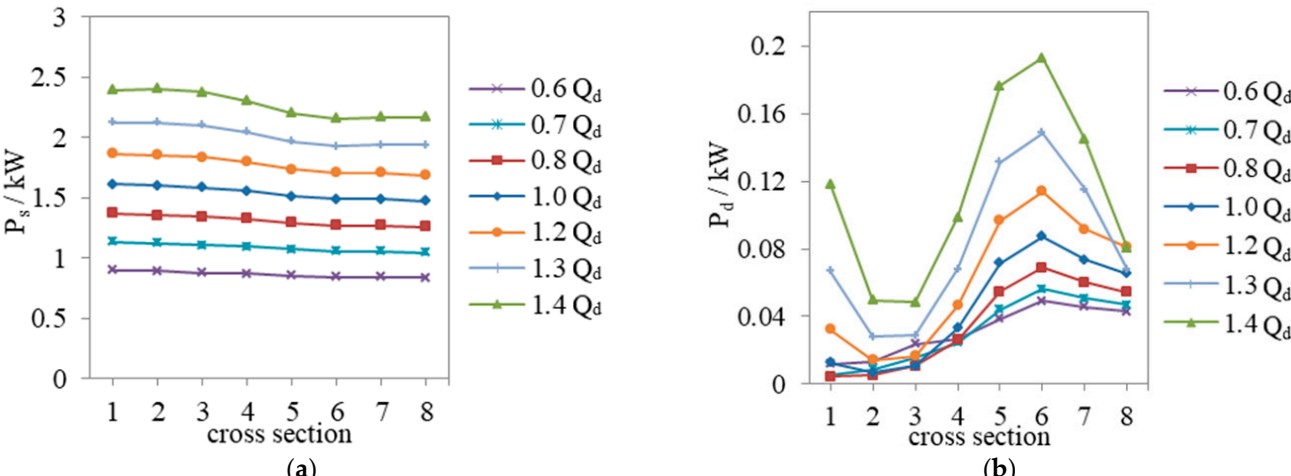

**Figure 9.** Static and dynamic pressure powers in each cross-section of the runner. (**a**) Static pressure power; (**b**) dynamic pressure power.

As shown in Figure 9, the variation tendency of static pressure power on each cross-section had a resemblance to that of the total pressure power, meaning the static pressure power from the runner inlet to outlet cross-section decreased sequentially and the amplitude difference of the static pressure power of the first four cross-sections was obviously greater than that of the last four cross-sections. In addition, the static pressure power of each cross-section was in proportion to the flow rate; in other words, the static pressure power of each cross-section increased with the growth of the flow rate and the static pressure power value of each cross-section was significantly greater than the dynamic pressure power value. There were three main reasons for static pressure power changes in the runner. The first was that the flow area for each cross-section from the runner inlet to outlet was different, resulting in a conversion between static and dynamic pressure powers when the fluid passed through each cross-section in turn. The second was that the static pressure energy does work on the runner. The third was the loss of partial static pressure energy overcoming hydraulic resistance.

It can also be seen from the Figure 9 that the relationship between the dynamic pressure power and the flow rate on all set cross-sections was different. The relationship between the dynamic pressure power and the flow rate on cross-sections 2, 3, and 4 was weak, but overall, the dynamic pressure powers on all set cross-sections had a positive correlation with the flow rate; in the other words, the larger the flow rate, the higher the dynamic pressure power on all cross-sections. This was because, when the flow area of each cross-section was constant, the flow rate was greater and the kinetic energy of the fluid increased correspondingly when the fluid passed through the cross-section.

Due to the hydraulic loss in the runner, the net energy input to the runner was unable to entirely transform into practicable energy. The mechanical energy obtained from the runner was measured by Equation (9), that is, the energy transferred from fluid to the runner. Figure 10 shows the distribution of energy transferred to the runner in each section under different flow rates.

$$P_{out} = M\omega \tag{9}$$

in which $M$ is the torque produced by the force of the runner shaft and $\omega$ is the runner angular velocity.

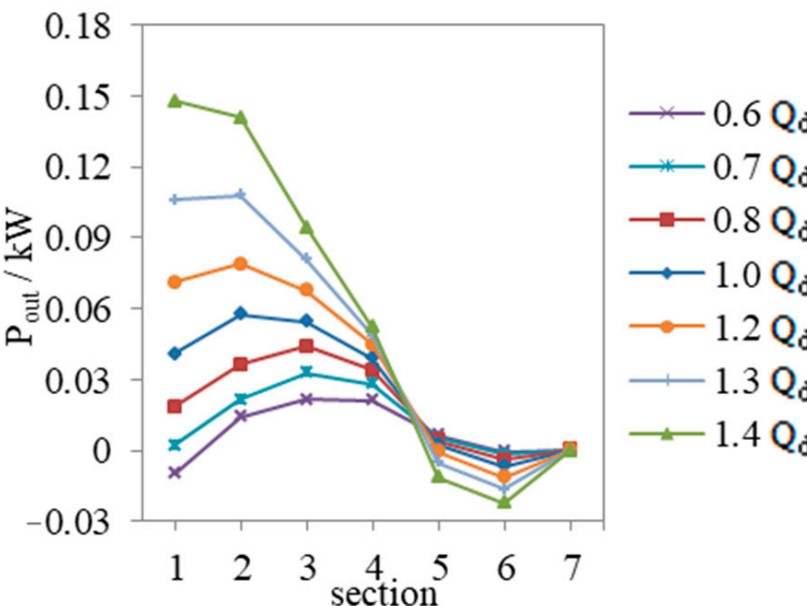

**Figure 10.** Output power of the runner in each section.

To further study the energy conversion characteristics in the runner, the variation characteristics of energy transferred to the runner in each section was analyzed. Figure 10 shows the variation curve of energy transferred to the runner in each section under different flow rates. It can be obtained that the energy received from the runner was primarily from the first four sections of the runner. The energy transferred from these four sections was in proportion to the flow rate. The first three sections of the runner obtained most of the energy, the obtained energy gradually increased as the flow rate increase, and the most energy-obtained section continuously moved from the third to the first section as the flow rate increased. For the fifth and sixth sections, the energy transferred to the runner was in inverse proportion to the flow rate. The energy obtained under small flow conditions was not much, and as the flow rate increased, these two sections not only could not obtain energy but had a counterproductive effect. Since the seventh section had no blade, the runner could hardly get the energy from this section, and the energy transferred to the runner had little correlation with the flow rate.

### 3.3. Detailed Flow Analysis and Discussion

Figure 11 is the power contour in the cross-section of the runner. It is shown that the power distribution characteristics of each cross-section of the runner under design flow rate were generally similar, with different details. In general, the power distribution of each cross-section was uneven, showing a trend of local concentration. At the inlet of the blade (Figure 11a), the energy was concentrated in the first and second flow passages facing the nozzle, and the extreme values appeared in this area. The maximum value appeared in the center of the nozzle jet, the minimum value appeared on the wall of the blade that was not directly impacted in the two flow channels, and the minimum value was a negative value. The negative power distribution was due to the flow separation in this area with high probability, and this flow separation was the secondary flow against the main flow direction. In the front half of the blade (Figure 11b–d), the energy distribution was basically the same as that at the inlet of the blade. The energy was still concentrated in the first and second flow passages facing the nozzle; however, the negative power in this area gradually decreased and disappeared along the flow direction, while the low energy area in the other flow passages always existed. In the latter half of the blade (Figure 11e,f), the energy distribution from the pressure to suction side of the blade was not uniform and the range of the low energy area was further reduced and gradually concentrated towards the top of

the blade. At the let of the blade (Figure 11g), the energy distribution showed a uniform trend in the circumferential direction.

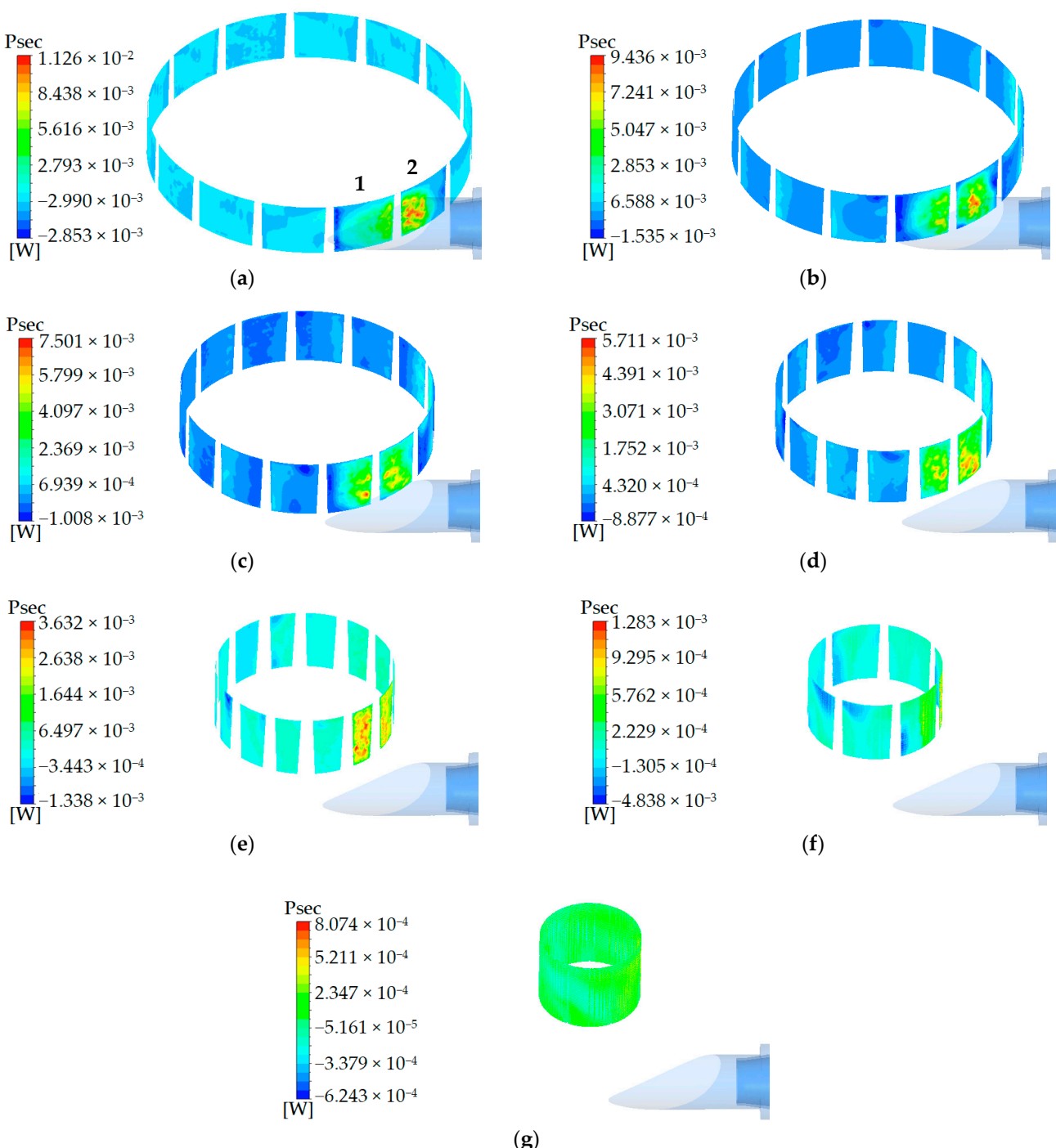

**Figure 11.** Power contour for cross-sections of the runner. (**a**) D = 175 mm; (**b**) D = 155 mm; (**c**) D = 135 mm; (**d**) D = 115 mm; (**e**) D = 95 mm; (**f**) D = 75 mm; (**g**) D = 55 mm.

Figure 12 shows the fluid energy dissipated in the shape of turbulent kinetic energy. It can be seen that, except for the flow passage impacted by the nozzle, there was basically no turbulent dissipation rate in the other flow channels and it remained unchanged. The turbulent dissipation rate in the flow passage impacted by the nozzle increased from the runner inlet along the flow direction and then slightly increased near the runner outlet, and the turbulent dissipation rate near the blade suction side was larger than that of the

pressure side. It is possible that the cause for this distribution characteristic of the turbulent dissipation rate was the narrow flow passage of the runner and the dissipation rate was greatly affected by the wall. Therefore, distributions of turbulent dissipation rate and velocity in the runner (as shown in Figure 13) show similar characteristics, and the magnitude close to the blade suction side was greater than that of the blade pressure side.

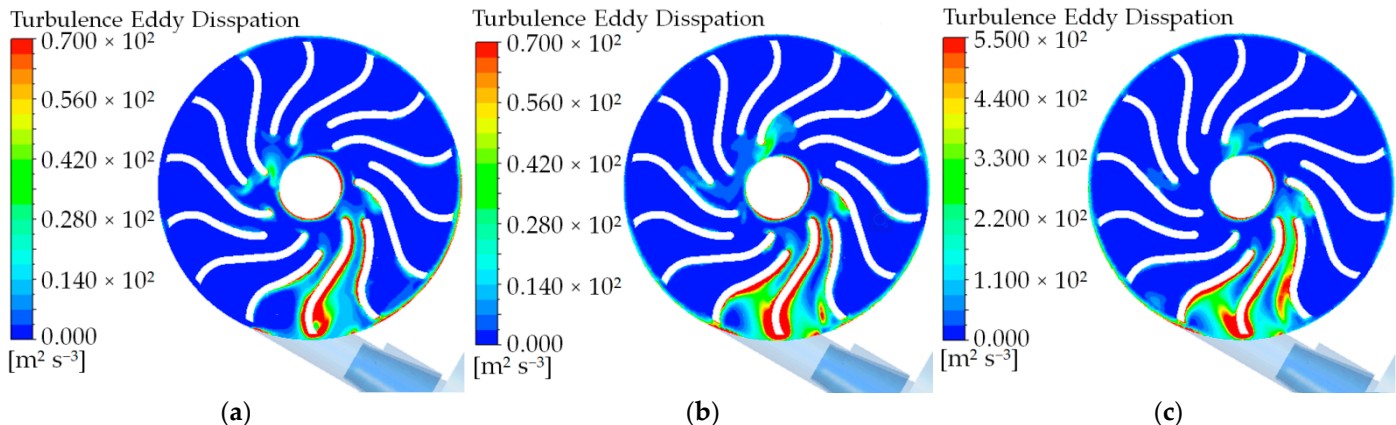

**Figure 12.** Turbulence eddy dissipation in the mid-span section of the runner. (**a**) $0.7Q_d$; (**b**) $1.0Q_d$; (**c**) $1.3Q_d$.

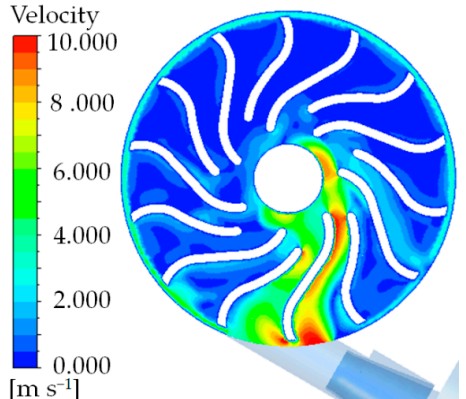

**Figure 13.** Velocity in the mid-span section of the runner.

## 4. Conclusions

The impulse water turbine with the splitter blade in this paper is a promising device for the utilization of environmentally friendly energy, the performance of which was investigated. On the basis of the above results and discussions, the following conclusions are deduced:

(1) The high-efficiency area of the turbine moves to the direction of a high flow rate as the rotational speed increases. The range of the high efficiency at high rotational speed is wider than that at a low rotational speed, and the peak efficiency is 42.93%. Therefore, this water turbine still has the potential to improve performance.

(2) The capacity of the runner to do work from the inlet to outlet gradually decreases, the front half of the blades is the main part to do work, and the capacity of the runner to do work gradually decreases with the raise of flow rate. The output of the runner shows similar performance to its work capacity. In addition, the back of the runner blades consumes the mechanical energy of the impeller and produces negative output as the flow rate increases.

(3) The high energy area for the runner is concentrated in the flow passage facing the nozzle, the energy is gradually evenly distributed from the runner inlet to the runner outlet, and the negative energy caused by flow separation with high probability is gradually reduced. In addition, the fluid energy dissipated in the form of turbulent kinetic energy can

also be observed in the flow passage facing the nozzle, and its distribution characteristics are likely related to the shape of the runner flow passage.

(4) Working efficiency of the front half of the blade significantly affects the working efficiency of the entire water turbine. Thus, the improvement of the runner structure, especially for the front half of the blade, will further increase the performance of the impulse water turbine.

**Author Contributions:** Conceptualization, L.T. and S.Y.; methodology, L.T. and Y.T.; software, L.T.; validation, L.T. and Y.T.; formal analysis, L.T., S.Y. and Y.T.; investigation, L.T. and S.Y.; resources, L.T. and Z.G.; data curation, Z.G.; writing—original draft preparation, L.T.; writing—review and editing, S.Y. and Y.T.; supervision, S.Y. and Y.T.; project administration, L.T.; funding acquisition, L.T. All authors have read and agreed to the published version of the manuscript.

**Funding:** This research was funded by the Natural Science Foundation of Jiangsu Province (Grants No. BK20180876), China Postdoctoral Science Foundation Funded Project (Project No. 2019M651735), and the Key Laboratory Project of Water-saving Irrigation Engineering, Ministry of Agriculture (Project No. FI2I2019-01-0101).

**Conflicts of Interest:** The authors declare no conflict of interest.

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
