# Peer review of "Performance Characteristics in Runner of an Impulse Water Turbine with Splitter Blade"

_processes, doi:10.3390/pr9020303_

Round 1
Reviewer 1 Report
The manuscript presents analyze of an impulse water turbine in order to detect the places where dissipated energy occurs during transformation of fluid energy into mechanical one.
Methodology of the research relays on analyze of power change, which reflect energy conversion. The research is based on CFD simulations and test of turbine on the experiment rig, in order to validate CFD methodology comparing main parameters of turbine obtained by the both method. Before main calculation there was the grid sensitivity investigated.
The proposals for improvements, observations and minor shortcomings, which are difficult to avoid, are shown below:
- The authors used for flow simulations a CFD software. Please inform readers, what type of software was applied?
- For simulation of flow through the turbine the authors applied k-ε turbulence model. I recommend to justify why this model was chosen instead other ones, e.g. k-ω model.
- In conclusion (4) the authors stated: “The improvement of the runner structure especially for the front half of the blade will further increase the performance of the impulse water turbine”. I recommend to justify: why the front half of blade should be improved?
- It seems that graphs in Figure 6. (a and b) are swapped each other.
- In the line no 228 is written that “cross section 7 is the runner outlet”, but in Figure 7. Is shown that “cross section 8” is runner outlet. Please improve the error.
- It would be good to add to the boundary conditions value of total pressure at the turbine outlet.
- In the Figure 1. the additional characters near the position numbers are not needed. Remove them, please.
Reviewer 2 Report
The paper presents CFD simulations and experiments of performance characteristics in runner of an impulse water turbine with splitter blade. I have the following comments to the paper.
- What kind of software was used for simulations? I presume a commercial software. This must be mentioned explicitly, otherwise the paper can not be accepted for publication.
- The proper choice of the k-eps turbulence model in comparison with several others must be also justified. Which of the k-eps models was actually used? The appropriate value of the y+ in the first cell next to the solid boundary depends on the model. How can a reader be sure that the selected k-eps turbulence model performs better than the others implemented in the CFD software used in simulations? I suggest the authors to take example from the paper in the and Transactions of the ASME. J. Turbomachinery, 2013, 135(1), paper 011027 and Transactions of the ASME. J. Heat Transfer, 2014, 136 (5) paper 051901, which deals in detail with these issues.
- The grid-independence study is not complete. All grids used in the study apparently operated with 10 prism layers. Usually, one must vary the number of prism layers. Exact values of the y+ in the first cell next to the solid boundary for each grid must be mentioned. In any case, grids with the boundary layer resolution of y+ close to unity must be also involved in the study in combination with the appropriate turbulence models (realizable k-eps, k-omega SST etc.). I suggest the authors to have a look at the papers mentioned above, as well as at the paper in Proc. 2008 ASME Turbo Expo (June 9-13, 2008, Berlin, Germany), Vol. 4, Pts. A and B, pp. 1051-1061 (ASME pap. GT2008-51207) to see, how a grid-independence study should be performed and described in full.
- That the selected turbulence model in combination with the selected grid perform worse than might be expected can be seen from Fig. 5. Simulations of the efficiency systematically overpredict the experiments for all values of the rotation speed in r.p.m.
- The language is overall good; an additional proofreading editing is, however, strongly recommended. Grammatic errors can be seen from time to time throughput the text.
- Once the suggested papers are used for the revision, they must be added to the references list.
Based on the said above, I suggest major revisions.
Round 2
Reviewer 2 Report
The revised paper has incorporated answers to all my comments. One minor thing needs to be corrected yet.
- The newly included Ref. [37] is incorrect, because it is an entirely experimental paper, which does not deal with the CFD issues such as the turbulence models, grid-independence, y+ etc. The correct reference is:
[37] Shevchuk, I.V., Jenkins, S.C., Weigand, B., von Wolfersdorf, J., Neumann, S.O., and Schnieder, M., “Validation and Analysis of Numerical Results for a Varying Aspect Ratio Two-Pass Internal Cooling Channel”, Proceedings of the 2008 ASME Turbo Expo; Vol. 4, Pts. A and B, pp. 1085-1094.
In my initial review I accidentally mentioned the incorrect page range, which caused this misunderstanding. Thant was a pure misprint.
Based on the said above, I suggest a minor revision.
